# Concise and Free-Metal Access to Lactone-Annelated Pyrrolo[2,1-*a*]isoquinoline Derivatives via a 1,2-Rearrangement Step

**DOI:** 10.3390/ijms25021085

**Published:** 2024-01-16

**Authors:** Arina Y. Obydennik, Alexander A. Titov, Anna V. Listratova, Tatiana N. Borisova, Victor B. Rybakov, Leonid G. Voskressensky, Alexey V. Varlamov

**Affiliations:** 1Organic Chemistry Department, Science Faculty, Peoples’ Friendship University of Russia (RUDN University), 6 Miklukho-Maklaya Street, Moscow 117198, Russia; arina.abydennik@gmail.com (A.Y.O.); titov-aa@rudn.ru (A.A.T.); listratova-av@rudn.ru (A.V.L.); borisova-tn@rudn.ru (T.N.B.); varlamov-av@rudn.ru (A.V.V.); 2Department of Chemistry, Lomonosov Moscow State University, Leninskie Gory, 1-3, Moscow 119991, Russia; rybakov20021@yandex.ru

**Keywords:** hexaflouroisopropanol, lactonic pyrrolo[2,1-*a*]isoquinolines, pyrido[2,1-*a*]isoquinolines, [1,2]-sigmatropic rearrangement, trifluoroethanol

## Abstract

Here, An efficient approach to obtaining previously unknown furo[2′,3′:2,3]pyrrolo[2,1-*a*]isoquinoline derivatives from readily available 1-R-1-ethynyl-2-vinylisoquinolines is described. The reaction features a simple procedure, occurs in hexaflouroisopropanol and does not require elevated temperatures. It has been found that the addition of glacial acetic acid significantly increases the yields of the target spirolactone products. Using trifluoroethanol instead of hexaflouroisopropanol results in the formation of pyrido[2,1-*a*]isoquinolines.

## 1. Introduction

The γ-lactone moiety is present in many bioactive natural products isolated from various plants and fungal metabolites [1,2,3]. Compounds with lactone and spirolactone fragments are characterized by a broad range of bioactivities and find their application in the field of medicine and agriculture. Thus, *trans*-dehydrocrotonin exhibits hypolipidemic and hypoglycaemic properties and has anti-cancer activity [4,5,6,7]; tetranorditerpenoids can be used as herbicides [8]; dehydroleucodine has anti-inflammatory and antiulcer activities [9]; and Stemoamide, Stemonamine and Tuberostemospironine, being *Stemona* alkaloids, possess anti-inflammatory, insecticidal, antitussive activities (Figure 1) [2,10,11]. Lactonic pyrrolizidinone alkaloids—pyrrolizilactone and UCS1025A—demonstrate potent antibacterial and antitumor effects [3,12].

Due to having a wide profile of pharmaceutical activities, spirolactones attract considerable attention from scientists and advance both the development of simple and effective synthetic routes to such structures and the further study of their properties. Recently, numerous methods for the synthesis of spirolactones have been described in the literature [13,14,15,16]. Among a variety of known approaches, those that are based on mild, free-metal and step-economic reactions start from readily available materials and meet the requirements of modern and “advantageous” synthetic chemistry, and so deserve special attention. Domino processes, incorporating the rearrangement and reconstruction of the carbon skeleton and leading to the complexity of a molecule’s structure to be quickly revealed in one step, can be considered as eligible candidates, fitting all the requirements of “advantageous chemistry” [17].

## 2. Results and Discussion

Herein, we report a study devoted to elucidating the divergent transformations of 1-R-1-ethynyl-2-vinyl-substituted 1,2,3,4-tetrahydroisoquinolines **1a**–**g** which occur in protic fluorinated solvents. One of the observed transformations proceeds via a 1,2-rearrangement step in the presence of AcOH/HFIP and opens up access to previously unexplored furo[2′,3′:2,3]pyrrolo[2,1-*a*]isoquinoline derivatives **3**.

Previously, we have described the chemical behavior of 1-R-1-ethynyl-2-vinyl-substituted 1,2,3,4-tetrahydroisoquinolines in aprotic solvents [18]. It has been shown that the route of the MW-stimulated rearrangements deeply depends on the type of solvent used. The use of toluene favored the formation of pyrrolo[2,1-*b*][3]benzazepines, while switching to acetonitrile afforded pyrido[2,1-*a*]isoquinolines in good yields. Encouraged by these unusual results, we decided to examine the influence of protic solvents, particularly fluorinated alcohols—trifluoroethanol and hexafluoroisopropanol (HFIP)—on the disclosed rearrangements. Fluorinated alcohols are characterized by having a low nucleophilicity and high ionizing and solvating power, increased Brønsted acidity in the hydroxyl proton and high polarity, as well as the ability to affect the regio- and chemoselectivity of a reaction and its process rate [19,20,21,22]. In other words, they could open up new directions for these well-known transformations.

The initial 1-R-1-ethynyl-2-vinyl-substituted 1,2,3,4-tetrahydroisoquinolines **1a**–**g** were obtained, according to a previously described procedure, from the corresponding 3,4-dihydroisoquinolines and methyl propiolate [18]. We began our study with transformations of tetrahydroisoquinolines **1a**–**g** so it would arise in less acidic trifluoroethanol (pKa = 12.4) [19]. To our delight, the conversions did not require elevated temperatures and proceeded smoothly at 20 °C to generate pyrido[2,1-*a*]isoquinolines, in 55–95% yields, as the sole product (Table 1). To understand what caused the change in the transformation route, the effect of the fluorinated alcohol or simply the presence of a protic solvent, we carried out a reaction of isoquinoline **1a** with a non-fluorinated ethanol (pKa = 15.9) [19]. Substrate **1a** was transformed into product **2a** but the use of ethanol as a solvent slowed down the process three times; in addition, the yield of the target compound decreased to 78%. We have already reported on the synthesis of pyrido[2,1-*a*]isoquinolines from isoquinolines **1a**–**f** in acetonitrile in the presence of triphenylphosphine [18]. In that case, the conversions required more severe conditions, which makes it less attractive compared to the present protocol.

Inspired by the results obtained in trifluoroethanol, we decided to explore the intramolecular changes when starting with tetrahydroisoquinolines **1a**–**g** in more acidic hexafluoroisopropanol (HFIP) (pKa = 9.3) [19]. Using isoquinoline **1a** as a model substrate, we performed a reaction at 20 °C. The transformation proceeded smoothly, but, to our surprise, led to a reaction mixture which consisted of lactonic pyrrolo[2,1-*a*]isoquinoline **3a** (25%) and pyrido[2,1-*a*]isoquinoline **2a** (71%) (Table 2, entry 1). The formation of **3a** was completely unexpected. The literature survey has not revealed the analogous structures, and we have succeeded only in finding isomeric ones [23]. It was clear that the acidity of the solvent played a key role. Given our earlier published studies demonstrating that the use of more acidic solvents such as HFIP and AcOH can alter the routes in the transformation of 1-R-ethynyl-decorated tetrahydroisoquinolines in reaction with activated alkynes towards more thermodynamically stable products, we considered that increasing the acidity of the medium with acetic acid would promote the construction of product **3a [24,25]**. Indeed, the yield of the desired **3a** was improved to 43% by adding 0.5 equiv of glacial acetic acid; however, the formation of compound **2a** was still observed (Table 2, entry 2). The best result was achieved with 3.0 equiv of AcOH to produce lactone **3a** with a 55% yield. It is noteworthy that a further increase in acetic acid did not have any significant effect on the yield of the target compound **3a** (Table 2, entries 3 and 4).

With the optimized conditions in hand, we investigated the scope of the discovered transformation. To estimate the effect of the substituents attached at C-1 during the intramolecular changes to tetrahydroisoquinolines **1b**–**g**, experiments with different alkyl and aryl substituents were carried out. Isoquinolines **1b**–**d** with isopropyl, benzyl and phenyl groups proved to be good substrates for the transformation, producing lactonic pyrrolo[2,1-*a*]isoquinolines **3b**–**d** in 50–64% yields (Figure 1). However, the presence of substituents in the phenyl radical at C-1 affected both the composition and the ratio of the reaction mixtures. Thus, isoquinolines **1e**–**f** containing electron-donating substituents (-OMe and -F) in the *para*-position in the phenyl ring provided pyrrolo[2,1-*b*][3]benzazepines **4** and **5**; no traces of lactones were observed (Figure 2). We have already published a paper describing the construction of the pyrrolo[2,1-*b*][3]azepines scaffold via [3,3]-sigmatropic rearrangement in vinyl- and ethynyl-substituted di(tetra)hydroisoquinolines [18], but again the present version of the reaction stood out due to its simplicity and mild reaction conditions. *para*-Nitrophenyl-substituted isoquinoline **1g** produced a mixture of products, consisting of pyrido[2,1-*a*]isoquinoline **2g** (47%), 1-ylidene pyrrolo[2,1-*a*]isoquinoline **6b** (19%) and lactone **3g** (13%) (Figure 1).

The structure of 1-ylidene pyrrolo[2,1-*a*]isoquinoline **6b** was assigned on the basis of NOESY, HMQC and HMBC spectra (Figure 2, see Appendix A). The NOESY spectrum has correlations between H-1 and H-3 in the pyrrole cycle as well as between H-1 and H-5 and H-10 in the isoquinoline moiety. In the HMBC spectrum, there are correlations between H-1 and C-1, C-3, C-10b in the pyrrole cycle; C-5, C-2 in the ester group; and C-6 in the aryl substituent.

Catalytic routes towards lactones where HFIP facilitates the formation of the products [26,27] which are known in the literature. We believe that the transformation commences with the HFIP-assisted polarization of the enamine moiety (Figure 3). The subsequent formation of the pyrrole ring (**A**) followed by the migration of a proton from the solvent to the anionic center of the ylidene fragment results in an intermediate (**B**). The following [1,3]-shift gives cation (**C**) in which a Wagner–Meerwein rearrangement occurs to furnish the intermediate (**D**). The final lactonization of the latter leads to the formation of the target products **3**.

The ambiguous behavior of isoquinolines in HFIP in the presence of 3.0 equiv of acetic acid returned us to the idea of carrying out these reactions without any additives. At 20 °C in HFIP, isoquinolines **1a**–**g** formed multicomponent mixtures, from which the products were isolated using column chromatography. As was expected in the case of the starting compounds **1b**–**d** with isopropyl, benzyl and phenyl substituents, the yields of lactones **3** decreased (Figure 1). But again, the isoquinolines **1e**–**g** decorated with *para*-OMe, *para*-F and *para*-NO_2_ phenyl radicals at C-1 stood out from the general scheme. Now, isoquinolines **1e**–**f** having electron-donating groups demonstrated the highest yields of the desired lactone **3**. The formation of lactone **3e** was accompanied by the formation of product **6a**—1-ylidene-substituted pyrrolo[2,1-*a*]isoquinolines—with a 15% yield (Figure 1). In the case of isoquinoline **1g** with the *para*-NO_2_ phenyl radical, we did not find the corresponding lactone **3g**; from the reaction mixture we obtained, pyrido[2,1-*a*]isoquinoline **2g** and 1-ylidene-pyrrolo[2,1-*a*]isoquinoline **6b** were isolated with 43% and 31% yields, respectively (Figure 1).

## 3. Materials and Methods

### 3.1. General Information

IR spectra were recorded on an Infralum FT-801 FTIR spectrometer on KBr tablets for crystalline compounds or on a film for amorphous compounds (ISP SB RAS, Novosibirsk, Russia). ^1^H and ^13^C NMR spectra were acquired on a 600 MHz NMR spectrometer (JEOL Ltd., Tokyo, Japan) from CDCl_3_ to acquire compounds with a solvent signal as the internal standard (7.27 ppm for ^1^H nuclei, 77.2 ppm for ^13^C nuclei); peak positions were given in parts per million (ppm, *δ*). Multiplicities are indicated by s (singlet), d (doublet), t (triplet), m (multiplet). Coupling constants, *J*, are reported in Hertz. HRMS spectra were recorded on an AB SCIEX TripleTOF 5600+ mass-spectrometer (AB Sciex Pte. Ltd., Singapore) using electrospray ionization (ESI). The measurements were conducted in a positive-ion-mode mass range from m/z 100 to 1000. A syringe injection was used for solutions in MeOH (concentration 100 ng/mL, flow rate 100 μL/min). Melting points were determined on SMP-10 apparatus (Bibby Sterilin Ltd., Stone, UK) with open capillary tubes. Sorbfil PTH-AF-A-UF plates (Imid Ltd., Krasnodar, Russia) were used for TLC; visualization was carried out in an iodine chamber or using KMnO_4_ and H_2_SO_4_ solutions. Silica gel (40–60 μm, 60 Å) from Macherey-Nagel GmbH&Co (Loughborough, UK) was used for column chromatography. All reagents (Sigma-Aldrich, St. Louis, MO, USA; Merck, Darmstadt, Germany; J.T. Baker, Phillipsburg, NJ, USA) were used without additional purification. Compounds **1a**–**f**, **2a**–**f** and **4** were also prepared earlier according to the described procedures [18].

Deposition Number 2156399 (for **3a**) contains the supplementary crystallographic data for this paper. These data are provided free of charge by the joint Cambridge Crystallographic Data Centre and Fachinformationszentrum Karlsruhe Access Structures service (see Appendix A).

### 3.2. General Procedure for the Synthesis of Compound ***1g***

Methyl propiolate (3.0 mmol) was added to the solution of corresponding isoquinoline (1.0 mmol) in 7 mL of CH_2_Cl_2_. The reaction was carried out at room temperature. The progress of the reaction was monitored by TLC (Sorbfil, EtOAc-hexane, 1:1). The solvent was removed under reduced pressure; in the case of compound **1g**, the residue was purified by column chromatography on silica gel (1:5 EtOAc–hexane).

**Methyl (2*E*)-3-[6,7-dimethoxy-1-(3-methoxy-3-oxoprop-1-yn-1-yl)-1-(4-nitrophenyl)-3,4-dihydroisoquinolin-2(1*H*)-yl]prop-2-enoate (1g).** Yield 0.397 g (83%), yellow oil. IR spectrum (KBr), υ/cm^−1^: 2231 (C≡C), 1717 (C=O), 1519, 1349 (NO_2_). ^1^H NMR (600 MHz, CDCl_3_) *δ* 8.23–8.21 (m, 2H, H-Ar), 7.68–7.66 (m, 2H, H-Ar), 7.36 (d, *J* = 13.6 Hz, 1H, -CH=CH-CO_2_Me), 6.65 (s, 1H, 8-CH), 6.39 (s, 1H, 5-CH), 4.94 (d, *J* = 13.6 Hz, 1H, -CH=CH-CO_2_Me), 3.88 (s, 3H, OCH_3_), 3.82 (s, 3H, OCH_3_), 3.67 (s, 3H, OCH_3_), 3.66–3.63 (m, 1H, 3-CH_2_), 3.62 (s, 3H, OCH_3_), 3.49–3.46 (m, 1H, 3-CH_2_), 3.09–3.05 (m, 1H, 4-CH_2_), 2.95–2.92 (m, 1H, 4-CH_2_). ^13^C NMR (150 MHz, CDCl_3_) *δ* 169.0, 153.4, 149.3, 149.0, 148.8, 148.5, 148.0, 128.6 (2C), 127.2, 125.6, 124.3 (2C), 111.2, 111.1, 92.8, 84.7, 80.5, 64.3, 56.2, 56.1, 53.2, 51.1, 42.8, 27.9. HRMS (ESI) m/z calc’d for C_25_H_24_N_2_O_8_ [M+H]⁺ 481.1605, found: 481.1605 (0.0 ppm).

### 3.3. General Procedure for the Synthesis of Compounds ***2a**–**g***

Isoquinolines **1a**–**g** (0.3 mmol) were dissolved in 2,2,2-trifluoroethanol (7 mL). The reaction was carried at room temperature. The progress of the reaction was monitored by TLC (Sorbfil, EtOAc-hexane, 1:1). The solvent was removed under reduced pressure, the residue was crystallized from Et_2_O to produce compounds **2a**, **2c**–**f**; in the case of compounds **2b** and **2g,** the residue was purified by column chromatography on silica gel (1:5 EtOAc–hexane). Yields of **2a**–**f** in 2,2,2-trifluoroethanol: **2a** (95%), **2b** (55%), **2c** (56%), **2d** (71%), **2e** (79%), **2f** (80%). The spectral data for compounds **2a**–**f** are similar to those previously obtained and reported in [18].

**Dimethyl 11b-(4-nitrophenyl)-9,10-dimethoxy-7,11b-dihydro-6*H*-pyrido[2,1-*a*]isoquinoline-2,3-dicarboxylate (2g).** Yield 0.098 g (68%), light yellow oil. IR spectrum (KBr), υ/cm^−1^: 1688 (C=O), 1519, 1347 (NO_2_). ^1^H NMR (600 MHz, CDCl_3_) *δ* 8.10 (d, *J* = 8.8 Hz, 2H, H-Ar), 7.81 (s, 1H, 4-CH), 7.54 (s, 1H, 1-CH), 7.27–7.25 (m, 2H, H-Ar), 7.01 (s, 1H, 11-CH), 6.67 (s, 1H, 8-CH), 3.90 (s, 3H, OCH_3_), 3.76 (s, 6H, 2*OCH_3_), 3.60–3.56 (m, 1H, 6-CH_2_), 3.39 (s, 3H, OCH_3_), 3.34–3.30 (m, 1H, 6-CH_2_), 3.00–2.96 (m, 1H, 7-CH_2_), 2.80–2.77 (m, 1H, 7-CH_2_). ^13^C NMR (150 MHz, CDCl_3_) *δ* 167.0, 164.3, 161.2, 156.0, 149.1, 147.8, 147.6, 147.2, 129.4 (2C), 126.2, 126.1, 123.1 (2C), 112.3, 111.3, 105.7, 104.1, 78.7, 56.1, 55.9, 51.1, 51.0, 42.4, 29.2. HRMS (ESI) m/z calc’d for C_25_H_24_N_2_O_8_ [M+Na]⁺ 503.1425, found: 503.1421 (−0.8 ppm).

### 3.4. General Procedure for the Synthesis of Compounds ***3a**–**g, 4, 5a,b*** and ***6a,b***

(A) Isoquinoline 1 (0.3 mmol) was dissolved in 7 mL HFIP. The reaction was carried out at room temperature. The progress of the reaction was monitored by TLC (Sorbfil, EtOAc-hexane, 1:1). The solvent was removed under reduced pressure, the residues were chromatographed on silica gel (1:3 EtOAc–hexane) to obtained compounds **3a**–**g** and **6a,b**.

(B) To a solution of isoquinoline **1** (0.3 mmol) in 7 mL HFIP, glacial AcOH (0.9 mmol) was added. The reaction was carried at room temperature. The progress of the reaction was monitored by TLC (Sorbfil, EtOAc-hexane, 1:1). The solvent was removed under reduced pressure; compounds **3a**–**g**, **4**, **5a,b** and **6b** were chromatographed on silica gel (1:5 EtOAc–hexane (for **4** and **6a,b**); 1:3 EtOAc–hexane (for **3a**–**g**, **5a,b**)).

**Methyl 10,11-dimethoxy-3a-methyl-2-oxo-3,3a,7,8-tetrahydro-2*H*-furo[2′,3′:2,3]pyrrolo[2,1-*a*]isoquinoline-4-carboxylate (3a).** Yield 0.059 g (55%), white solid, mp 210–212 °C. IR spectrum (KBr), υ/cm^−1^: 1764, 1680 (C=O). ^1^H NMR (600 MHz, CDCl_3_) *δ* 7.19 (s, 1H, 5-CH), 6.69 (s, 1H, H-Ar), 6.60 (s, 1H, H-Ar), 3.90 (s, 3H, OCH_3_), 3.87 (s, 3H, OCH_3_), 3.70 (s, 3H, OCH_3_), 3.68–3.60 (m, 2H, 7-CH_2_), 3.52 (d, *J* = 18.2 Hz, 1H, 3-CH_2_), 2.90 (d, *J* = 18.2 Hz, 1H, 3-CH_2_), 2.90–2.85 (m, 1H, 8-CH_2_), 2.74–2.70 (m, 1H, 8-CH_2_), 1.03 (s, 3H, CH_3_). ^13^C NMR (150 MHz, CDCl_3_) *δ* 174.9, 164.6, 150.0, 148.3, 146.5, 129.0, 122.4, 111.5, 109.2, 108.4, 104.9, 56.4, 56.0, 54.1, 50.8, 42.8, 40.5, 29.8, 21.6. HRMS (ESI) m/z calc’d for C₁₉H_2_₁NO_6_ [M+H]⁺ 360.1442, found: 360.1451 (2.5 ppm).

**Methyl 10,11-dimethoxy-2-oxo-3a-(propan-2-yl)-3,3a,7,8-tetrahydro-2*H*-furo[2′,3′:2,3]pyrrolo[2,1-*a*]isoquinoline-4-carboxylate (3b).** Yield 0.081 g (50%), white solid, mp 237–239 °C. IR spectrum (KBr), υ/cm^−1^: 1751, 1675 (C=O). ^1^H NMR (600 MHz, CDCl_3_) *δ* 7.35 (s, 1H, 5-CH), 6.67 (s, 1H, H-Ar), 6.62 (s, 1H, H-Ar), 3.90 (s, 3H, OCH_3_), 3.88 (s, 3H, OCH_3_), 3.79 (d, *J* = 17.9 Hz, 1H, 3-CH_2_), 3.68 (s, 3H, OCH_3_), 3.66–3.64 (m, 2H, 7-CH_2_), 2.98–2.93 (m, 1H, 8-CH_2_), 2.95 (d, *J* = 17.9 Hz, 1H, 3-CH_2_), 2.75–2.71 (m, 1H, 8-CH_2_), 1.85–1.79 (m, 1H, CH(CH_3_)_2_), 0.98 (d, *J* = 6.7 Hz, 3H, CH(CH_3_)_2_), 0.43 (d, *J* = 6.7 Hz, 3H, CH(CH_3_)_2_). ^13^C NMR (150 MHz, CDCl_3_) *δ* 174.6, 165.3, 150.1, 148.7, 148.1, 128.9, 122.4, 111.5, 109.7, 105.1, 102.2, 60.8, 56.4, 56.0, 50.7, 42.4, 39.7, 34.3, 29.3, 20.5, 16.4. HRMS (ESI) m/z calc’d for C_2_₁H_25_NO_6_ [M+H]⁺ 388.1755, found: 388.1765 (2.6 ppm).

**Methyl 3a-benzyl-10,11-dimethoxy-2-oxo-3,3a,7,8-tetrahydro-2*H*-furo[2′,3′:2,3]pyrrolo[2,1-*a*]isoquinoline-4-carboxylate (3c).** Yield 0.083 g (64%), white solid, mp 218–220 °C. IR spectrum (KBr), υ/cm^−1^: 1762, 1676 (C=O). ^1^H NMR (600 MHz, CDCl_3_) *δ* 7.07 (t, *J* = 7.6 Hz, 1H, H-Ph), 7.04 (s, 1H, 5-CH), 6.95 (t, *J* = 7.6 Hz, 2H, H-Ph), 6.71 (s, 1H, H-Ar), 6.59 (s, 1H, H-Ar), 6.25 (d, *J* = 7.6 Hz, 2H, H-Ph), 3.95 (s, 3H, OCH_3_), 3.93 (s, 3H, OCH_3_), 3.78 (s, 3H, OCH_3_), 3.65 (d, *J* = 18.2 Hz, 1H, 3-CH_2_), 3.46–3.42 (m, 1H, 7-CH_2_), 3.31 (d, *J* = 14.1 Hz, 1H, -CH_2_-Ph), 3.30–3.27 (m, 1H, 7-CH_2_), 3.07 (d, *J* = 18.2 Hz, 1H, 3-CH_2_), 2.65 (d, *J* = 14.1 Hz, 1H, -CH_2_-Ph), 2.38–2.34 (m, 1H, 8-CH_2_), 1.85–1.80 (m, 1H, 8-CH_2_). ^13^C NMR (150 MHz, CDCl_3_) *δ* 174.3, 165.1, 150.5, 148.6, 147.8, 135.6, 130.3, 130.2 (2C), 127.3 (2C), 126.5, 122.5, 111.5, 109.4, 104.7, 104.1, 58.0, 56.6, 56.3, 51.0, 42.3, 41.2, 39.3, 28.9. HRMS (ESI) m/z calc’d for C_25_H_25_NO_6_ [M+H]⁺ 436.1755, found: 436.1757 (0.5 ppm).

**Methyl 10,11-dimethoxy-2-oxo-3a-phenyl-3,3a,7,8-tetrahydro-2*H*-furo[2′,3′:2,3]pyrrolo[2,1-*a*]isoquinoline-4-carboxylate (3d).** Yield 0.077 g (61%), white solid, mp 212–214 °C. IR spectrum (KBr), υ/cm^−1^: 1759, 1679 (C=O). ^1^H NMR (600 MHz, CDCl_3_) *δ* 7.37 (s, 1H, 5-CH), 7.10–7.08 (m, 2H, H-Ph), 7.06–7.04 (m, 1H, H-Ph), 7.02 (d, *J* = 7.6 Hz, 2H, H-Ph), 6.53 (s, 1H, H-Ar), 6.14 (s, 1H, H-Ar), 3.85–3.81 (m, 1H, 7-CH_2_), 3.79 (s, 3H, OCH_3_), 3.78 (br. D, *J* = 5.0 Hz, 2H, 3-CH_2_), 3.75–3.73 (m, 1H, 7-CH_2_), 3.57 (s, 3H, OCH_3_), 3.54 (s, 3H, OCH_3_), 3.04–3.00 (m, 1H, 8-CH_2_), 2.82–2.79 (m, 1H, 8-CH_2_). ^13^C NMR (150 MHz, CDCl_3_) *δ* 174.1, 164.1, 149.6, 147.7, 146.9, 138.7, 128.3 (3C), 127.5, 126.4 (2C), 122.7, 110.9, 110.6, 109.4, 106.1, 60.8, 56.0, 55.9, 50.8, 42.4, 38.0, 29.1. HRMS (ESI) m/z calc’d for C_24_H_23_NO_6_ [M+H]⁺ 422.1598, found: 422.1604 (1.4 ppm).

**Methyl 10,11-dimethoxy-3a-(4-methoxyphenyl)-2-oxo-3,3a,7,8-tetrahydro-2*H*-furo[2′,3′:2,3]pyrrolo[2,1-*a*]isoquinoline-4-carboxylate (3e).** Yield 0.067 g (50%), light yellow solid, mp 196–198 °C. IR spectrum (KBr), υ/cm^−1^: 1760, 1675 (C=O). ^1^H NMR (600 MHz, CDCl_3_) *δ* 7.34 (s, 1H, 5-CH), 6.92 (d, *J* = 8.6 Hz, 2H, H-Ar), 6.62 (d, *J* = 8.6 Hz, 2H, H-Ar), 6.53 (s, 1H, H-Ar), 6.18 (s, 1H, H-Ar), 3.84–3.80 (m, 1H, 7-CH_2_), 3.81 (s, 3H, OCH_3_), 3.75 (br. S, 2H, 3-CH_2_), 3.73-3.71 (m, 1H, 7-CH_2_), 3.69 (s, 3H, OCH_3_), 3.58 (s, 6H, 2*OCH_3_), 3.03–2.98 (m, 1H, 8-CH_2_), 2.81–2.78 (m, 1H, 8-CH_2_). ^13^C NMR (150 MHz, CDCl_3_) *δ* 174.3, 164.2, 158.6, 149.6, 147.7, 146.6, 130.7, 128.3, 127.5 (2C), 122.7, 113.6 (2C), 110.9, 110.6, 109.4, 105.9, 60.4, 56.0, 55.9, 55.2, 50.8, 42.4, 38.2, 29.1. HRMS (ESI) m/z calc’d for C_25_H_25_NO_7_ [M+H]⁺ 452.1704, found: 452.1714 (2.2 ppm).

**Methyl 3ª-(4-fluorophenyl)-10,11-dimethoxy-2-oxo-3,3ª,7,8-tetrahydro-2*H*-furo[2′,3′:2,3]pyrrolo[2,1-*a*]isoquinoline-4-carboxylate (3f).** Yield 0.080 g (61%), white solid, mp 206–208 °C. IR spectrum (KBr), υ/cm^−1^: 1771, 1683 (C=O). ^1^H NMR (600 MHz, CDCl_3_) *δ* 7.35 (s, 1H, 5-CH), 6.99–6.97 (m, 2H, H-Ar), 6.80–6.77 (m, 2H, H-Ar), 6.54 (s, 1H, H-Ar), 6.14 (s, 1H, H-Ar), 3.83–3.80 (m, 1H, 7-CH_2_), 3.81 (s, 3H, OCH_3_), 3.76 (d, *J* = 15.7 Hz, 2H, 3-CH_2_), 3.75–3.71 (m, 1H, 7-CH_2_), 3.58 (s, 6H, 2*OCH_3_), 3.03–2.98 (m, 1H, 8-CH_2_), 2.82–2.79 (m, 1H, 8-CH_2_). ^13^C NMR (150 MHz, CDCl_3_) *δ* 173.8, 164.1, 161.8 (d, *J* = 247.1 Hz, 1C), 149.7, 147.9, 146.9, 134.6, 128.4, 128.1 (d, *J* = 8.1 Hz, 2C), 122.4, 115.2 (d, *J* = 21.6 Hz, 2C), 111.0, 110.4, 109.3, 105.8, 60.4, 56.0, 55.9, 50.8, 42.4, 38.2, 29.1. HRMS (ESI) m/z calc’d for C_24_H_22_FNO_6_ [M+H]⁺ 440.1504, found: 440.1500 (−0.9 ppm).

**Methyl 10,11-dimethoxy-3ª-(4-nitrophenyl)-2-oxo-3,3ª,7,8-tetrahydro-2*H*-furo[2′,3′:2,3]pyrrolo[2,1-*a*]isoquinoline-4-carboxylate (3g).** Yield 0.018 g (13%), yellow solid, mp 147–149 °C. IR spectrum (KBr), υ/cm^−1^: 1774, 1682 (C=O), 1519, 1347 (NO_2_). ^1^H NMR (600 MHz, CDCl_3_) *δ* 7.98 (d, *J* = 8.8 Hz, 2H, H-Ar), 7.43 (s, 1H, 5-CH), 7.22 (d, *J* = 8.8 Hz, 2H, H-Ar), 6.58 (s, 1H, H-Ar), 6.12 (s, 1H, H-Ar), 3.90–3.83 (m, 3H, 3-CH_2_, 7-CH_2_), 3.81 (s, 3H, OCH_3_), 3.78 (d, *J* = 17.4 Hz, 1H, 3-CH_2_), 3.58 (s, 3H, OCH_3_), 3.56 (s, 3H, OCH_3_), 3.09–3.04 (m, 1H, 8-CH_2_), 2.89–2.85 (m, 1H, 8-CH_2_). ^13^C NMR (150 MHz, CDCl_3_) *δ* 172.8, 163.7, 150.4, 148.1, 147.2, 147.0, 146.1, 128.8, 127.5 (2C), 123.5 (2C), 121.6, 111.3 (2C), 108.8 (2C), 60.8, 56.1, 56.0, 51.1, 42.5, 38.0, 29.0. HRMS (ESI) m/z calc’d for C_24_H_22_N_2_O_8_ [M+H]⁺ 467.1449, found: 467.1455 (1.3 ppm).

**Dimethyl 11-hydroxy-8,9-dimethoxy-11-(4-methoxyphenyl)-6,11-dihydro-5*H*-pyrrolo[2,1-*b*][3]benzazepine-1,2-dicarboxylate (5a).** Yield 0.079 g (55%), orange oil. IR spectrum (KBr), υ/cm^−1^: 3521 (OH), 1723, 1709 (C=O). ^1^H NMR (600 MHz, CDCl_3_) *δ* 7.61 (s, 1H, 3-CH), 7.17 (s, 1H, 10-CH), 6.98 (d, *J* = 8.9 Hz, 2H, H-Ar), 6.77 (d, *J* = 8.9 Hz, 2H, H-Ar), 6.60 (s, 1H, 7-CH), 4.03–3.99 (m, 1H, 5-CH_2_), 3.93 (s, 3H, OCH_3_), 3.90 (s, 3H, OCH_3_), 3.89 (s, 3H, OCH_3_), 3.88–3.84 (m, 1H, 5-CH_2_), 3.80 (s, 3H, OCH_3_), 3.76 (s, 3H, OCH_3_), 3.64 (s, 1H, OH), 2.98–2.94 (m, 1H, 6-CH_2_), 2.86–2.82 (m, 1H, 6-CH_2_). ^13^C NMR (150 MHz, CDCl_3_) *δ* 169.6, 164.1, 159.6, 148.3, 147.6, 138.5, 137.2, 134.0, 128.4 (2C), 127.4, 127.2, 116.9, 114.0 (2C), 113.4, 113.3, 110.7, 77.7, 56.2, 56.1, 55.4, 52.8, 51.6, 48.2, 33.2. HRMS (ESI) m/z calc’d for C_26_H_27_NO_8_ [M+Na]⁺ 504.1629, found: 504.1641 (2.4 ppm).

**Dimethyl 11-(4-fluorophenyl)-11-hydroxy-8,9-dimethoxy-6,11-dihydro-5*H*-pyrrolo[2,1-*b*][3]benzazepine-1,2-dicarboxylate (5b).** Yield 0.062 g (44%), orange oil. IR spectrum (KBr), υ/cm^−1^: 3449 (OH), 1715 (C=O). ^1^H NMR (600 MHz, CDCl_3_) *δ* 7.59 (s, 1H, 3-CH), 7.17 (s, 1H, 10-CH), 7.06–7.03 (m, 2H, H-Ar), 6.94–6.91 (m, 2H, H-Ar), 6.61 (s, 1H, 7-CH), 4.02–3.98 (m, 1H, 5-CH_2_), 3.93 (s, 3H, OCH_3_), 3.90 (s, 3H, OCH_3_), 3.89 (s, 3H, OCH_3_), 3.87–3.85 (m, 1H, 5-CH_2_), 3.80 (s, 3H, OCH_3_), 3.75 (s, 1H, OH), 2.95–2.91 (m, 1H, 6-CH_2_), 2.87–2.83 (m, 1H, 6-CH_2_). ^13^C NMR (150 MHz, CDCl_3_) *δ* 169.6, 164.0, 162.6 (d, *J* = 248.5 Hz, 1C), 148.5, 147.7, 142.2, 136.9, 133.7, 129.1 (d, *J* = 8.1 Hz, 2C), 127.5, 127.4, 117.0, 115.6 (d, *J* = 21.6 Hz, 2C), 113.6, 113.4, 110.6, 77.6, 56.2, 56.1, 52.9, 51.6, 48.4, 33.2. HRMS (ESI) m/z calc’d for C_25_H_24_FNO_7_ [M+Na]⁺ 492.1429, found: 492.1434 (1.0 ppm).

**Methyl (1*E*)-8,9-dimethoxy-1-(2-methoxy-2-oxoethylidene)-10b-(4-methoxyphenyl)-1,5,6,10b-tetrahydropyrrolo[2,1-*a*]isoquinoline-2-carboxylate (6a).** Yield 0.021 g (15%), beige solid, mp 227–229 °C. IR spectrum (KBr), υ/cm^−1^: 1721, 1679 (C=O). ^1^H NMR (600 MHz, CDCl_3_) *δ* 7.31 (s, 1H, 3-CH), 7.21 (d, *J* = 8.8 Hz, 2H, H-Ar), 6.84 (d, *J* = 8.8 Hz, 2H, H-Ar), 6.60 (s, 1H, H-Ar), 6.42 (s, 1H, H-Ar), 5.60 (s, 1H, =CH-CO_2_Me), 3.90 (s, 3H, OCH_3_), 3.82 (s, 3H, OCH_3_), 3.79 (s, 3H, OCH_3_), 3.75 (s, 3H, OCH_3_), 3.74–3.70 (m, 1H, 5-CH_2_), 3.69 (s, 3H, OCH_3_), 3.33–3.30 (m, 1H, 5-CH_2_), 3.12–3.08 (m, 1H, 6-CH_2_), 2.69–2.66 (m, 1H, 6-CH_2_). ^13^C NMR (150 MHz, CDCl_3_) *δ* 169.5, 165.7, 159.5, 148.4, 148.3, 146.8, 136.9, 130.6, 129.8 (2C), 129.3, 126.3, 120.0, 113.7 (2C), 111.4, 109.3, 94.4, 64.7, 56.2, 56.1, 55.4, 52.4, 51.0, 48.0, 28.8. HRMS (ESI) m/z calc’d for C_26_H_27_NO_7_ [M+H]⁺ 466.1860, found: 466.1861 (0.2 ppm).

**Methyl (1*E*)-8,9-dimethoxy-1-(2-methoxy-2-oxoethylidene)-10b-(4-nitrophenyl)-1,5,6,10b-tetrahydropyrrolo[2,1-*a*]isoquinoline-2-carboxylate (6b).** Yield 0.045 g (31%), orange oil. IR spectrum (KBr), υ/cm^−1^: 1733, 1699 (C=O), 1518, 1349 (NO_2_). ^1^H NMR (600 MHz, CDCl_3_) *δ* 8.19–8.17 (m, 2H, H-Ar), 7.51–7.49 (m, 2H, H-Ar), 7.32 (s, 1H, 3-CH), 6.65 (s, 1H, H-Ar), 6.35 (s, 1H, H-Ar), 5.59 (s, 1H, =CH-CO_2_Me), 3.91 (s, 3H, OCH_3_), 3.83 (s, 3H, OCH_3_), 3.75 (s, 3H, OCH_3_), 3.71 (s, 3H, OCH_3_), 3.68–3.64 (m, 1H, 5-CH_2_), 3.41–3.39 (m, 1H, 5-CH_2_), 3.16–3.11 (m, 1H, 6-CH_2_), 2.73–2.70 (m, 1H, 6-CH_2_). ^13^C NMR (150 MHz, CDCl_3_) *δ* 169.0, 165.3, 150.2, 148.9, 148.7, 147.6, 146.8, 130.7, 129.3 (2C), 129.1, 126.4, 123.8 (2C), 118.8, 111.7, 109.0, 95.9, 64.6, 56.3, 56.1, 52.6, 51.2, 48.2, 28.5. HRMS (ESI) m/z calc’d for C_25_H_24_N_2_O_8_ [M+H]⁺ 481.1605, found: 481.1605 (0.0 ppm).

## 4. Conclusions

In summary, we have described a novel procedure for the synthesis of lactonic pyrrolo[2,1-*a*]isoquinolines and pyrido[2,1-*a*]isoquinolines through the rearrangements of 1-R-1-ethynyl-2-vinyl-1,2,3,4-tetrahydroisoquinolines in fluorinated alcohols. It has been demonstrated that the rearrangements depend on the acidity of the solvents used. In some cases, the addition of 3 equiv of AcOH increased the yields of the target lactones. The substituent at C-1 in the starting isoquinolines affects the composition and the ratio of the products in the transformation occurring in HFIP both with and without AcOH.

## Data Availability

The data presented in this study are available in the article and in the Appendix A.

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
