# Peer review of "Concise and Free-Metal Access to Lactone-Annelated Pyrrolo[2,1-a]isoquinoline Derivatives via a 1,2-Rearrangement Step"

_ijms, 2024, doi:10.3390/ijms25021085_

Round 1

Reviewer 1 Report

Comments and Suggestions for Authors

The manuscript titled “A concise and free-metal access to lactone annelated pyrrolo[2,1-a]isoquinoline derivatives via a 1,2-rearrangement step” reported a newly protocol for achieving lactonic pyrrolo[2,1-a]isoquinolines and pyrido[2,1-a]isoquinolines from 1-R-1-ethynyl-2-vinylisoquinolines. The developed synthetic method is commendable for its the acceptable scope and procedural simplicity, along with the notable usage of TFE/HFIP as solvent at ambient temperatures. The authors elucidated the structures of the formed products under conditions with single crystal X-ray diffraction technology, NOESY, HMQC and HMBC. Additionally, a plausible mechanism was proposed based on the literature. Overall, after minor revisions, this work could be considered for publication in a high-quality journal as International Journal of Molecular Sciences.

1. As described in the manuscript, “Fluorinated alcohols are characterized by low nucleophilicity, high ionizing and solvating power, increased Brønsted acidity of the hydroxyl proton, high polarity as well as the ability to affect the regio- and chemoselectivity of a reaction and its process rate [19,20].” It appears that two important and pioneering literature (Regio- and Stereoselective Ring-Opening Reactions of Epoxides with Indolesand Pyrroles in 2,2,2-Trifluoroethanol. Chem. Eur. J. 2008,14, 1638–1647.; Friedel–Crafts alkylation of arenes with epoxides promoted by fluorinated alcohols or water. Chem. Commun., 2010, 46, 2653–2655.) have been missed. These two studies represent pioneering research on reactions promoted by fluorinated alcohol, showcasing notable achievements in terms of regio- and chemoselectivity. The authors are suggested to cite these two important papers.

2. As described in the manuscript, “The structure of 1-ylidene pyrrolo[2,1-a]isoquinoline 6b was assigned on the basis of NOESY, HMQC, and HMBC spectra (Figure 2). The NOESY spectrum has correlations of H-1 to H-3 of the pyrrole cycle as well as its correlation to H-5 and H-10 of the isoquinoline moiety. In the HMBC spectrum there are correlations of H-1 to C-1, C-3, C-10b of pyrrole cycle; to C-5, C-2 of the ester group and to C-6 of the aryl substituent.” It seems challenging for readers to quickly grasp the key information mentioned in Figure 2. The authors suggested to enhance clarity by labeling the information/results revealed by the NOESY/HMQC/HMBC with different colors. This visual aid would assist readers in swiftly identifying the crucial details rather than the spare ones.

3. As mentioned “1H and 13C NMR spectra were acquired on 600-MHz NMR spectrometer (JEOL Ltd., Tokyo, Japan) in CDCl3 for compounds with a solvent signal as internal standard (7.27 ppm for 1Н nuclei, 77.2 ppm for 13С nuclei);”. It looks that the authors took 77.04/77.05/77.11 ppm for 13С nuclei) in the spectra provided in the supporting information (Page 3-15 in the supporting information). The authors should rectify this.

4. As mentioned in the section of “3. Materials and Methods”, “B) To a solution of isoquinoline 1 (0.3 mmol) in 7 ml HFIP glacial AcOH (0.9 mmol) was added.” should be corrected as “B) To a solution of isoquinoline 1 (0.3 mmol) in 7 ml HFIP, glacial AcOH (0.9 mmol) was added.”

5. The details (solvent used, temperature et al) of the preparation of the single crystal of 3a should be included.

Author Response

Dear Reviewer,

Thank you for the thorough examination of our paper. We followed all the suggestions made and implemented the following changes:

  1. As it was recommended two publications, demonstrating pioneering research on reactions promoted by fluorinated alcohol, have been added.
  2. We have marked with different colors the important information in the description of spectra.
  3. Chemical shifts of CDCl3 signals as internal standard (77.2 ppm for 13С nuclei) have been corrected.
  4. We agree with the recommendation and the corresponding correction has been done.
  5. The details related to the preparation of monocrystals of compound 3a have been added.

Sincerely yours,

Dr. Alexander A. Titov,

Organic Chemistry Department,

Peoples’ Friendship University of Russia (RUDN University),

6, Miklukho-Maklaya St., Moscow.

Reviewer 2 Report

Comments and Suggestions for Authors

This manuscript reports a metal-free synthesis of lactone annelated pyrrolo[2,1-a]isoquinoline derivatives. The results showcase the formation of new heterocyclic motif. Few major issues found are

1. Introduction is very brief. It should include some more relevant details on the similar methods/biological activities.

2. Compounds 2a-g are formed in trifluoroethanol, however, increasing acidity to HFIP, a different nucleus is obtained. Some examples are made with the addition of AcOH while some are only formed in HFIP, how authors justify the difference?  

3. Several typo errors exists such as change one of the observed transformation to one of the observed transformations.

4. Authors should use appropriate notations such as degrees and other symbols.

5. The substrate scope is limited. few more examples could be added based on the reactivity and outcome of the list examples as some of them are giving mixture of products.

Comments on the Quality of English Language

Minor editing is required.

Author Response

Dear Reviewer,

Thank you for the thorough examination of our paper. We followed all the suggestions made and implemented the following changes:

  1. In the literature there are a limited number of bioactive compounds having γ-lactone moiety annelated with nitrogen-containing heterocycles (pyrrole, pyridine, azepine and ect.). Basically, they belong to a big family of Stemona alkaloids. We considered that it was not necessary to cite all structures with lactone fragment and decided to limit ourselves to mentioning some of them, which corresponds to the type of the manuscript – “Communication”. The synthesis of similar system with lactonic pyrroloisoquinoline framework is presented only in one publication, which is cited in our work, but it is multi-step.
  2. We have shown that increasing the acidity (as well as solvating power along with low nucleophilicity) of fluorinated alcohols led to the formation of lactonic pyrroloisoquinolines from enyne substituted isoquinolines. Except for 1-(para-nitrophenyl) derivative, the revealed conditions are suitable for the other corresponding starting isoquinolines. The addition of AcOH (proton donor) affects the transition of intermediate A to B thus increasing the yields of the products – annelated lactones, and only in the case of the isoquinolines with electron-donating groups in para-position of phenyl radical the regioselectivity of the process was changed. Under these conditions isoquinolines with electron-donating groups in para-position of phenyl radical were transformed into pyrrolobenzazepines.
  3. We have corrected typo errors.
  4. Notations have been checked and corrected.
  5. We agree that now the number of substrate scope is limited. We used only basic/classical examples of starting enyne substituted isoquinolines with alkyl and aryl radicals containing electron-donating and electron-withdrawing/halogen groups. As the synthesized scaffold is unique, unstudied and promising, we continue to investigate the substrate scope. Based on the results of molecular docking we are planning to obtain new compounds with other substituents, as well as to conduct bioactive tests. All the future results will be published in due course.

Sincerely yours,

Dr. Alexander A. Titov,

Organic Chemistry Department,

Peoples’ Friendship University of Russia (RUDN University),

6, Miklukho-Maklaya St., Moscow.

Round 2

Reviewer 2 Report

Comments and Suggestions for Authors

Authors have addressed the comments. The manuscript can be considered in the present form.

Comments on the Quality of English Language

Minor editing required.